# Low-Frequency Noise: Experiences from a Low-Frequency Noise Perceiving Population

**DOI:** 10.3390/ijerph20053916

**Published:** 2023-02-22

**Authors:** Kristina H. Erdélyi, Anselm B. M. Fuermaier, Lara Tucha, Oliver Tucha, Janneke Koerts

**Affiliations:** 1Department of Clinical and Developmental Neuropsychology, Faculty of Behavioral and Social Sciences, University of Groningen, 9712 TS Groningen, The Netherlands; 2Department of Psychiatry and Psychotherapy, University Medical Center Rostock, 18147 Rostock, Germany; 3Department of Psychology, National University of Ireland, W23 F2H6 Maynooth, Ireland

**Keywords:** low-frequency noise, LFN, perceptions, complaints, demographic characteristics

## Abstract

Although low-frequency noise (LFN) is associated with various complaints, there is still much unknown about this phenomenon. This research aims to provide an extensive description of (1) LFN perceptions, (2) LFN-related complaints, and (3) the characteristics of LFN complainants. In an explorative observational cross-sectional survey study, a sample of Dutch adults reporting to experience LFN (n = 190) and a group not experiencing LFN (n = 371) completed a comprehensive questionnaire. Descriptions of LFN perceptions varied individually and were dependent on different circumstances, although some common patterns were observed. Complaints were wide-ranging and individual, with a reported high impact on daily living. Common complaints included sleeping difficulties, fatigue, or annoyance. Societal consequences were described regarding housing, work, and relationships. Attempts to stop or escape the perception were manifold but often unsuccessful. The LFN sample differed regarding sex, education level, and age from the Dutch adult population, indicating more frequent inability to work, less full-time work, and less years lived in their homes. No further differences in occupational or marital status or living circumstances were found. Although this research supports some previous findings and identifies common patterns, it also highlights the individual nature of LFN-related experiences and the heterogeneity of this group. It is advised to pay attention to the complaints of affected individuals, to inform concerned authorities, and to conduct more systematic and multidisciplinary research using standardized and validated measuring instruments.

## 1. Introduction

Noise pollution currently constitutes the third largest environmental pollutant in Europe [1] affecting about a quarter of the European population [2]. It was associated with various adverse health effects including cardiovascular disease, sleep disturbance, annoyance, and cognitive impairment [3]. In contrast to general noise with no specification to its frequencies, the so called low-frequency noise (LFN) consisting of low frequency sounds around and below the human hearing threshold, is still a poorly understood and recognized environmental stressor. The Dutch Institute for Public Health and the Environment defines LFN as sound at frequencies between 20 Hz and 100/125 Hz, and sound below 20 Hz as infrasound [4]. However, some other definitions use the cut-offs of 200 Hz [5] to up to 250 Hz [6]. LFN is predominantly produced by man-made sources, such as ventilation systems, traffic, or turbines. With the rapidly growing industrialization, the number of LFN sources and LFN complaints is also rising [4,7]. According to the Dutch Institute for Public Health and the Environment (RIVM), more LFN reports have been received by health authorities since 2016 than reports of normal noise, and according to Bengtsson and Waye [8], on average, 44% of all noise complaints to health authorities represent LFN complaints.

While LFN can be audible at high levels (high volume), much of LFN is emitted below the average human hearing threshold [9], and therefore is not consciously perceived by the general population. Specifically, the RIVM estimates that about 8% of the Dutch adult population suffers some annoyance and 2% suffer severe annoyance from LFN [4]. A descriptive meta-analysis by Baliatsas et al. [10] estimated the pooled prevalence of high annoyance due to LFN at about 10.5% with prevalence ranging between 2% and 34%. Still, it is currently unknown what proportion of the population is perceiving LFN and why some individuals are more sensitive than others.

The conscious perception of LFN might depend on individual differences in hearing thresholds and sensitivity to specific frequencies [5], meaning that an individual may perceive a sound of specific frequencies with already very low sound levels. Such noise sensitivity, amongst others, constitutes a substantial predictor for noise reactions and psychological health outcomes [11]. When considering environmental research, in general, it was suggested that objectively measured environmental exposures do not always correspond with perceived ones [12]. Further, environmental noise research suggests, that while noise exposure is indeed a relevant predictor for noise reactions, especially annoyance, its strength varies based on the specific noise source and can only partially explain reactions to noise [13]. Specifically, it was estimated that noise exposure could only account for up to one third of the variance in annoyance, and another third of the variance could be explained by non-acoustic factors [14]. Various research in recent decades has highlighted the fragmentary role of noise exposure as a single predictor [15,16] and emphasized the relevance of non-acoustic factors including: (1) personal factors, such as attitudes, expectations, noise sensitivity, personality, coping, and demographics, (2) contextual and situational factors, such as the degree of urbanization, the visibility and predictability of the noise, and (3) social aspects of noise management, such as satisfaction with noise management procedures or the sound insulation [11,14,17,18,19,20,21,22]. Such factors can play a crucial role for expectations towards the presence of noise, the noise perception itself, annoyance judgement, symptom reporting, and general community responses, and they can be relevant determinants for health equity and environmental justice. Even the WHO Environmental Noise Guidelines for the European Region [23] highlight personal variables and situation factors as relevant moderators for residential noise annoyance judgements for road traffic noise. Considering that various noise sources included in environmental noise research can also be LFN sources (e.g., aircrafts or wind turbines), it is probable that such non-acoustic factors play a similar role in relation to LFN. However, knowledge about non-acoustic factors related to LFN specifically as well as differential characteristics of individuals reporting LFN perceptions is scarce [10].

In the pursuit of understanding LFN perceptions, a substantial heterogeneity regarding the time, location, and type of LFN perceptions needs to be considered. Regarding the time when LFN is perceived, an early survey examining LFN perceptions by Vasudevan and Gordon [24] suggested that LFN is more audible during the night. Similarly, Leventhall [5] found that LFN was most frequently perceived at night only (48%) followed by perceptions of LFN all the time (30%). Only small proportions (<10%) reported LFN perceptions at other specified times or with other circumstances. In contrast to those findings, in the annual report of a LFN volunteer organization, only 31% reported LFN perceptions at night only, and 65% reported perceiving it during both day and night. Only 4% reported daytime perception only [7]. Regarding the place of LFN perception, LFN seems to be mainly perceived in quiet rural and suburban regions [24], indoors rather than outdoors [24], and specifically at home (75% by [5]; 85% by [7]; 82% by [25]).

Considering the type of LFN perception, individuals most commonly reported hearing LFN (83% by [5]; 93% by [25]), but sometimes reported it as pressure in the ears or chest (41% in ears and 19% in the chest by [7]; [25]) or body vibrations (2% by [5]; 38% by [7]; 44% by [25]). Audibly perceived LFN was mostly described as humming (40% by [5]; 81% by [7]; [25]), and sometimes as buzzing (36% by [7]), a sound resembling an engine (22% by [5]; 40% by [7]; [25]) or throbbing and pulsing (22% by [5]; 12% by [7,24]). In conclusion, there seems to be considerable individual differences in when, where, and how LFN is perceived. A thorough investigation of LFN perceptions with the consideration of possible subgroups was not yet conducted.

Especially among individuals perceiving LFN, its exposure can elicit various adverse effects on health and functioning [5,7,26]. Considering the WHO definition of health, “a state of complete physical, mental, and social well-being and not merely the absence of disease or infirmity” [27], LFN seems to be associated with negative health outcomes by all of these factors. Specifically, three successive reviews on LFN in general [5] and the health effects of LFN [12,28] indicated complaints in physical, psychological, cognitive, and daily life functioning. Physical symptoms included cardiovascular complaints, heart palpitation, hypertension, nausea, body vibrations, and pain or pressure on the head or ears. Psychological symptoms included sleep disturbance, fatigue, annoyance, stress, nervousness, distraction, irritability, depression, or anxiety. Considering LFN-specific annoyance, LFN seems to be perceived as even more annoying and seems to cause annoyance at more silent levels than regular noise [6]. Cognition was shown to be negatively affected in attention, memory, and verbal as well as visual tasks. However, cognitive research does not yet allow for firm conclusions, with some studies suggesting no or even better cognitive performance during LFN exposure [29]. Finally, LFN seems to elicit substantial effects on daily living in terms of interference with everyday functioning, work performance, housing, social relations, increased drug use, and reported feelings of helplessness and frustration [5,7,10,29].

When investigating the sociodemographic factors of individuals reporting LFN complaints, first insights from mainly small scaled or pilot studies, surveys, or reports from volunteer organizations suggest that LFN perceptions are most commonly reported by individuals aged between 50 and 70 years [5,7,24,30,31]. Furthermore, LFN perceptions are in about two-thirds of cases reported by females [5,7,30,31], and LFN seems to affect cognitive performance more prominently in women [32].

All these above-described research findings suggest that more attention has to be paid to LFN as a serious environmental stressor as also emphasized by the WHO [33]. Although it was suggested that the effects of LFN on health seem to be more severe than the ones from general noise [6,28,33], systematic research on the individuals reporting LFN perceptions and their LFN-related experiences and complaints is scarce. Current knowledge is often based on studies of small groups or specific settings (e.g., occupational noise effects). Another line of research uses short-term LFN exposure in laboratories or focuses on on-site measurements, however it focuses mainly on annoyance as an outcome measure and/or used restricted test batteries in terms of psychological or cognitive outcomes. A further limitation of this line of research is that LFN sound measurements can be costly and time-consuming, and that standard noise measurements (such as the use of A-weighting) are not sensitive to detect LFN components efficiently. In many instances, an external source of LFN complaints cannot or can only be partially found, making it even more difficult for affected individuals to receive attention and support for their experienced complaints. In order to pay attention to the subjective perceptions and complaints of individuals experiencing hindrance attributed to LFN perceptions, this research aims to explore and to provide a broad and extensive overview of the research questions: What are (1) the perceptions, (2) the complaints, and (3) the characteristics of individuals reporting to experience LFN in their daily life in contrast to individuals reporting not to experience LFN. Specifically, this study aims to conduct an explorative and descriptive questionnaire study from a behavioral and social science viewpoint without the use of noise measurements, noise indicators (such as L_den_, L_dn_, or L_aeq_), or exposure-response relationship analyses. An investigation of the subjective experiences of individuals reporting LFN perceptions can direct towards factors and subgroups to be assessed by future causal and correlational research, provide more specific recommendations for LFN interventions, and inform authorities, stakeholders, healthcare providers, and affected individuals. For this, this study obtained a sample of Dutch individuals reporting LFN perceptions and complaints in their daily life and a comparison sample not perceiving LFN in their daily living. Participants completed a self-report questionnaire that was composed for the assessment of (1) the time, location, influencing circumstances, and type of LFN perceptions, (2) the reported physical, psychological, social, and societal health-related complaints, the authorities and experts consulted regarding LFN perceptions and complaints, and participants’ medication intake, and finally, (3) demographic characteristics.

## 2. Materials and Methods

### 2.1. Procedure and Participants

An online call for participants was set up through the newsletter and homepage of the Stichting Laagfrequentgeluid (www.laagfrequentgeluid.nl, accessed on 12 January 2023) and individuals could indicate their interest to participate via email. The Stichting Laagfrequentgeluid is a Dutch volunteer organization aiming to inform about LFN and support individuals suffering from LFN. Inclusion criteria required participants to currently perceive LFN and LFN-related difficulties in their daily life, to be 18 years or older, and to have a good command of Dutch. LFN perceptions were based on subjective reports without an on-site noise exposure measurement in order to receive a broad sample of people with LFN complaints independent of the success of determining a source. Participants received an information letter with inclusion and exclusion criteria as well as the current definition of LFN provided by the Dutch Institute for Public Health and the Environment (>125 Hz). Yet, since infrasound can be audible at appropriate levels [34] with potential effects on health [10], and since definitions of the upper cut-off of LFN are varying, individuals that reported measured frequencies at their dwellings in the infrasound range or above 125 Hz (but still under 250 Hz) were not excluded. Further possible LFN exposure indicators, although not utilized as in-or exclusion criteria, included an investigation of participants urbanization level. On request, participants recruited via the Stichting Laagfrequentgeluid or snowball sampling received a paper-pencil version of a set of questionnaires, including an informed consent and a form to indicate interest to participate in a further, on-site LFN study. No financial reward was provided. For the LFN group, out of 306 initially interested participants, 200 questionnaires were received between June 2018 and February 2021, resulting in a response rate of 65%. Most often, participants did not provide a reason for not sending back the battery of questionnaires. Withdrawal reasons included time constraints or hope for on-site measurements.

Among those 200, five participants were excluded because they did not report current perceptions of LFN. Further, five participants with significant neurological (e.g., epilepsy) and psychotic disorders (e.g., schizophrenia) were excluded because of assumed confounding effects of sound perceptions, as well as psychological and cognitive impairments not related to LFN. Participants with neurological (n = 6, 3%) or psychiatric disorders (n = 33, 17%) presumed to have a low confounding effect on the outcome variables were retained in the data set. Most often, participants reported a diagnosis of depression (7%) or anxiety disorder (4%). Participants with a professional diagnosis of tinnitus (n = 38, 20%) were not excluded from the research due to the difficulty to separate tinnitus patients from individuals perceiving LFN, and the possibility of comorbidity between those two conditions [5]. Furthermore, 9% were assumed to have tinnitus although not having received a diagnosis (n = 17). The final sample entering data analysis (n = 190) consisted of 188 Dutch adult residents in the Netherlands (99%) and two in Belgium (1%).

A comparison group (CG) of Dutch adults from the general population was recruited via the Dutch research panel “PanelInzicht”, an online platform offering financial compensation for participation in online studies. The CG received the same questionnaire and informed consent online and was sampled to have similar distributions in terms of sex (male, female), age categories (in the categories 18–34, 35–49, and 50–87 years), and education categories (“low”, “middle”, and “high”). The classification of education level is based on the Dutch educational system. Low education refers to the Dutch lagere school, LBO, VMBO basis, VMBO kader, VMBO-gl, LTS, or LEAO degrees. Middle education refers to the MBO, VMBO-t, MULO, MAVO, MTS, MEAO, HAVO, or Atheneum/Gymnasium degrees. Finally, high education refers to the HBO, HEAO, Pabo, HTS, university bachelor, university master, or higher degrees. Individuals with neurological or psychiatric disorders or a diagnosis of tinnitus were excluded from participation. From initial 723 participants, participants reporting to experience LFN-related complaints “regularly”, “often”, or “continuously” (n = 127) were excluded, as well as participants rating the extent of LFN-related restrictions in their daily life as a three or higher on a scale from 1 (not at all) to 10 (very much) (n = 124). Accordingly, participants reporting LFN-related complaints to occur “never” or “sometimes” were included, and participants that rated LFN-related restrictions as a one or two. Because the distribution of demographic characteristics has changed due to these exclusions, further 101 participants were excluded to obtain a comparison group with similar demographic characteristics as described above. Eventually, from 371 comparison participants, 364 (98%) lived in the Netherlands, 6 (2%) in Belgium, and one in Ireland (0.3%). As an additional comparison group for the analysis of demographic characteristics, openly available data from the general Dutch population (DP) was used from January 2021 or dates as close to January 2021 as possible.

The current study presents the first part of a larger project on LFN-related complaints. Ethical approval for the research was obtained from the Ethical Committee of Psychology (ECP) affiliated with the University of Groningen, the Netherlands (Registry nr. 17255, PSY-1819-S-0165, PSY-2122-S-004).

### 2.2. Materials

Being part of a larger research project, this study focuses on self-reported demographic characteristics, perceptions of LFN, and LFN-related complaints, based on a comprehensive inventory developed by the researchers. The selection of questions and answer options was primarily based on (1) the questions proposed to assess humming by the Dutch National Institute for Public Health and the Environment [30]. Further sources were (2) questions used by other LFN survey studies and common answers to them [25,35], (3) questions used to assess LFN complaints by the Stichting Laagfrequentgeluid (https://www.laagfrequentgeluid.nl/wordpress/meldingsformulier/, accessed on 25 May 2018) and common answers to them [7], and (4) other literature regarding LFN-related experiences and complaints (such as [9]). This questionnaire used closed questions (CQ), open questions (OQ), and a mixture of closed and open questions where participants could choose between various options but might also add another option or comment to the question (CQ/OQ). Further, it contained short answer questions (SA) requiring the participant to fill in a 1–2 word answer and one Likert scale question (LS) assessing the perceived extent of experienced complaints from 1 (not at all) to 10 (very much). Filling out this questionnaire required about 15 min. The LFN group received the complete questionnaire. The CG answered only the questions about demographic characteristics, medication intake, and the frequency and extent of experienced LFN complaints with the provision of a definition of LFN. Additionally, the CG also reported the frequency and extent of complaints due to general noise. For an overview of measured constructs, please refer to Table 1, and for the complete questionnaire, to Appendix A.

Regarding demographic data from the Dutch population, openly available data from the Dutch national statistical office (CBS) were used. For the analysis of housing type, the EU statistics on income and living conditions survey data (EU-SILC) were used.

### 2.3. Data Processing and Statistical Analysis

Overall, this study aims to provide a descriptive, broad overview of reported LFN-related experiences and complaints. For analyzing the first two research questions, the LFN-related (1) perceptions and (2) complaints, descriptive statistics of frequencies, proportions, distribution, central tendency, and dispersion will be used as well as bar charts to present quantitative data. Regarding the gathered qualitative data, a content analysis was conducted on the provided open answers within the coding program Atlas.ti Ver.9. Specifically, open answers were either placed into pre-existing options defined by the questionnaire, when applicable, or placed into newly formed categories, if possible. Answers selected by less than 10% of the participants and not fitting into other overarching categories were counted within an overarching “Other” category. Answers selected by two or less participants that were not fitting to any category were regarded as “individual answers”. Responses that were inconclusive or did not provide a clear answer to the question were disregarded. For some questions, the pre-existing answers from the questionnaire and the open answers were combined into a newly formed system, which described the given answers more accurately. This was the case for questions about the type of perceived sound, the type of LFN perception, the location of LFN perception, the assumed source, and the actions and their success in reducing nuisance from LFN.

#### 2.3.1. Medication Usage

Participants reported the name, dose, and intake frequency of their medication. Prescription-free medication was included, but vitamins, food supplements, contraceptives, or alternative forms of therapy were excluded from the analysis. Named medications were grouped into three categories: (1) calming medication (i.e., sleep medication and antidepressant/antipsychotic medication), (2) cardiovascular medication, and (3) other medication (i.e., skin, respiratory, pain, or gastrointestinal medication). The number of named medications was counted as well as the number of medication types that could either be calming, cardiovascular, or a subcategory of other medication (i.e., skin, respiratory, pain, or gastrointestinal medication).

#### 2.3.2. Demographic Characteristics

In order to describe the demographic characteristics, descriptive statistics of distribution, central tendency, and dispersion will be used. Additionally, group comparisons were conducted by means of hypothesis testing at a significance level of 0.05. For continuous variables, the nonparametric Mann–Whitney U test was used since the assumption of normality was violated. Normality was checked through the skewness and kurtosis values, the Shapiro–Wilk test, as well as visually by boxplots, histograms, and Q-Q plots. Homogeneity of variance was tested with Levene’s test. For categorical variables, chi-square tests were used, or Fisher’s z test when the expected count of a cell was less than 5. The magnitude of group differences was estimated by the effect size measure of Cohen’s r for continuous data and interpreted as small (0.1 < r < 0.3), medium (0.3 ≤ r < 0.5), or large (r ≥ 0.5) [36]. For categorical data, Cramer’s V or Cramers Phi for the Fisher’s z test was used based on Cohen [36]. For 1 degree of freedom, effect sizes were regarded as small (0.1), medium (0.3), and large (0.5), for 2 degrees of freedom, small (0.07), medium (0.21), and large (0.35), and for 5 degrees of freedom, small (0.04), medium (0.13), and large (0.22).

Further, participants were asked for a short answer on their occupation. Answers were analyzed for persons who are currently working or are work eligible. The latter includes individuals who are currently incapacitated, on sick leave or unemployed, and who have a job they identify with. Not considered were individuals ineligible for work or whose daily living status is best described by something else other than work including pension, student, or homemaker. Multiple jobs in different occupational fields were possible and counted. Answers were classified into 12 occupational groups based on the Dutch occupational classification system ROA-CBS 2014 derived from the International Standard Classification of Occupations 2008 (ISCO 2008).

Based on participant’s postal codes, the degree of urbanization of participants’ living locations was categorized based on the CBS definition of: extremely urbanized 2500 addresses or more per km^2^, strongly urbanized; 1500 to 2000 addresses per km^2^, moderately urbanized; 1000 to 1500 addresses per km^2^, hardly urbanized; 500 to 1000 addresses per km^2^, not urbanized; fewer than 500 addresses per km^2^. In order to investigate the relationship between participants’ urbanization level and LFN complaints, Spearman’s rank correlations were conducted between urbanization and the frequency and extent of experienced LFN complaints. Correlations were interpreted as low: r = 0.10, medium/moderate r = 0.30 and strong/high r = 0.50 [36]. Geographical distributions of participants’ living place based on postal code areas, urbanization levels, and the frequency and extent of LFN complaints were created with the program ArcGISPro 3.0.

## 3. Results

Results will be presented by the provision of a short summary in the following section and comprehensive tables and figures in the Appendix A providing further details.

### 3.1. LFN Perceptions

#### 3.1.1. Date of First LFN Perception

Among the participants reporting LFN perceptions (n = 190), 93% reported a date on which they started perceiving the LFN (n = 177). Dates ranged between 1994 and 2020 with an exponentially growing pattern (see Figure 1) and a median of 2015 (*M*: 2013.01, *SD*: 5.13). The pattern shows a drop in 2018, however, most data (n = 164) was collected in 2018. Explicitly, 66 participants (35%) attributed the appearance of LFN to a specific day (with 26 reporting the first of the month), 63 (33%) to a specific month, 37 (20%) to a specific year, and 12 could only approximate the first perception to a year. Regarding the occurrence of LFN at a specific month, no clear pattern was observed (see Appendix A). However, significantly more (z = −2.20 *p* = 0.03) participants reported the first occurrence in autumn/winter (n = 77, 60%) compared to spring/summer (n = 52, 40%), as also depicted in Appendix A.

#### 3.1.2. Type of LFN Perception

A detailed overview of the types and numbers of reported LFN perceptions is provided in Appendix A. Almost all participants reported perceiving LFN auditorily (n = 170, 90%), but also 81% of the participants reported feeling LFN (n = 153). Interestingly, 102 participants (54%) reported perceiving vibrations, among which 94% reported perceiving both, vibrations together with an auditory LFN sound.

Considering sound perceptions, participants reported perceiving between one and seven different sound types (*M* = 2.7, *Mdn* = 2.0, *SD* = 1.5). Specifically, 26% reported to perceive one, 27% two, 21% three, and 27% four or more different sound types (see Appendix A). Sounds that could be either constant or non-constant were reported by most participants (n = 156, 82%), including humming (73%), droning (27%), and buzzing (23%). A dynamically changing sound was reported by 111 participants (58%), especially in the form of a sound resembling a thumping engine (53%), but also including other descriptions such as thumping or pulsation. Other, more individual sound type descriptions were provided by 13 individuals (7%), reporting, for example, a non-tonal sound or a dual tone.

In terms of felt LFN, most commonly a perception of a feeling was reported in specific body parts (n = 142, 75%). This included the head area (n = 111, 59%), mostly in the form of pressure on the ears (n = 99, 52%), but also body parts around the upper body/torso area (n = 89, 47%) or the limbs (n = 46, 24%). Additionally, about 12% reported perceptions in their whole body (n = 22), and 5% feeling it with those body parts that touch the ground or furniture (n = 9). Participants providing an additional description of the type of sensation named amongst other vibrations, pressure, or a resonating feeling.

#### 3.1.3. Location of LFN Perception

All participants reported to perceive LFN at inside locations (n = 190) and 74% (n = 140) reported perceiving LFN at outside locations. Specifically, 27% of the participants (n = 51) reported perceiving LFN (almost) everywhere. Another 26% reported to perceive it exclusively in inside locations (n = 50), either perceiving it only in one’s own home (14%, n = 26) or at multiple inside locations (13%, n = 24). The remaining 47% (n = 89) perceived LFN at both inside and outside locations, however, not everywhere. Descriptions could range from perceiving LFN within a limited radius around one´s home to all inside locations with some specific outside locations. For more details, please refer to Appendix A.

Notably, the reported locations were highly individual and could encompass very specific descriptions, such as one’s bedroom, workspace, caravan, garden, or places bound to a specific LFN source. Furthermore, various participants reported the intensity and type of LFN to differ per location, for example, LFN being perceived as more intense in inside locations compared to outside locations.

#### 3.1.4. Measurement and Frequency of Measured LFN

Two-thirds of the participants (66%, n = 126) reported that a measurement of the LFN was conducted or requested. About half of those participants reported that the LFN could be measured or perceived (n = 70, 37%), although nine participants specified that the measurement was conducted by themselves. From those 70 individuals, 51 reported a specific measured frequency (n = 26, 14%) or a frequency range/multiple frequencies (n = 25, 13%). Reported frequencies varied individually between 5 Hz and 140 Hz for the specific measured frequencies and between 0.02 and 150 Hz for the frequency range/multiple frequencies. From the 51 individuals reporting a frequency, the majority (n = 44, 23%) reported a frequency in the LFN range. Notably, 17 individuals (9%) reported a frequency (range) categorized as infrasound (<20 Hz) and two individuals (1%) reported a frequency (range) above what the Dutch guidelines consider as LFN (<125 Hz) (Appendix A). Interestingly, 50 Hz or a frequency range including 50 Hz was reported by multiple participants. For more details, please refer to Appendix A.

#### 3.1.5. Assumed Source of the LFN

One-third (33%, n = 62) of the participants did not know any source of the LFN, 43% (n = 82) provided an assumed source and the remaining 24% (n = 45) stated that they do not know a source for sure, but still provided assumptions. Among the participants that referred to a source, between one and nine different sources (*M* = 2.1, *Mdn* = 2.0. *SD* = 1.5) were named. Specifically, 49% reported one, 22% two, 13% three, and 17% four or more different sources (Appendix A).

Assumptions about the sources were highly individual. Most participants named air-conditioning and ventilation as a LFN source (n = 36, 20%), followed by machinery and household appliances (n = 33, 17%), pumps and water transport (n = 30, 16%), electrical installations (n = 29, 15%), traffic (n = 25, 13%), heating (n = 22, 12%), gas extraction and transport (n = 19, 10%), or other less frequently named sources (n = 45, 24%). Notably, 5% assumed the LFN to come from their neighbors. For a more detailed overview, please refer to Appendix A.

#### 3.1.6. Circumstances Influencing the LFN Perception

About a quarter of the participants reported to always perceive LFN in the same manner (24%, n = 45), while about three-quarters (76%, n = 144) reported special circumstances influencing their LFN perception. Most frequently, LFN was reported to depend on the time of the day (n = 81, 43%; see Appendix A), being especially present throughout the night (35%) and in the evening (22%). Another commonly reported circumstance was the presence of other sounds (n = 76, 40%), described most often as a masking effect by other sounds or noise (31%). Other influences included wind strength or direction (n = 30, 16%), the season (n = 28, 15%), the day of the week (n = 28, 15%), and other, less frequently named factors such as weather and temperature. For more details, please refer to Appendix A. Specifically, the descriptions of the effect of wind, day of the week, or temperature were highly heterogeneous and did not show an explicit pattern. Noticeably, 10% specifically named weekends and holidays to be associated with different LFN perceptions compared to workdays. Regarding the effect of season specifically, a tendency of autumn and winter to be associated with more frequent or more intense LFN perceptions (n = 22, 79%) was observed compared to spring and summer (n = 9, 32%; see Appendix A). Descriptions of weather effects suggested that humidity was associated with more frequent or more intense LFN perceptions (n = 9, 56%). A varying LFN perception, yet without a clearly identifiable pattern, was reported by 17% of the participants (n = 33). It has to be noted, however, that the above-mentioned and other factors are partly intertwined with each other.

#### 3.1.7. Other Persons Perceiving the Sound

About two-thirds of the participants (67%, n = 128) reported that there are other persons in their environment that perceive the LFN as well, while one-third (32%, n = 61) reported that there are no other persons perceiving the sound. Among the individuals reporting another person to perceive the LFN, between one and five different other persons were named (*M* = 1.8, *Mdn* = 2.0, *SD* = 0.94) with most participants naming one (n = 62, 49%; Appendix A). Most often, the participant’s partner (n = 53, 28%) was reported, followed by guests (n = 38, 20%), neighbors (n = 37, 20%), or a family member (n = 35, 19%). Other, less frequently named persons were housemates, friends, locals, researchers/measurement employees, colleagues, and other acquaintances. For more details, please refer to Appendix A. Notably, 8% of the participants stated that the other person perceived LFN less frequently/loudly or experienced it as less bothering.

#### 3.1.8. Actions to Reduce Nuisance Due to LFN and Their Success

Almost all participants (n = 183; 96 %) tried various actions to reduce LFN nuisance. Specifically, between 1 and 15 different actions were tried with an average of 4.5 different actions (*Mdn* = 5, *SD* = 1.8) and with half of the participants (n = 94, 50%) naming 4 or 5 tried measures (Appendix A). Most commonly, participants tried to reduce the LFN (n = 165, 87%) by using earplugs/earphones (n = 163, 86%) and in a few cases through noise-canceling headphones or placing something in or on their ears. The majority of participants (n = 155, 82%) also tried to mask the sound. This was done mostly via the TV or radio (n = 145, 76%), and, in fewer instances, through other sounds such as white noise or ventilation. Furthermore, 75% of the participants (n = 143) tried to eliminate or switch off a suspected source, and also 75% (n = 143) tried to change or adapt their living location predominantly through closing or opening windows (n = 142, 75%), and in fewer cases through insulating walls or windows or going to another location. Finally, about 57% (n = 108) tried to change or adapt their sleeping location, mainly through repositioning their bed (n = 107, 56%), and few participants also reported sleeping at a different location such as one’s caravan, garden, or kitchen floor, placing their bed on dampening materials, or changing their mattress. Other, less frequently named approaches involved, amongst others, contacting authorities and neighbors, distraction, coping, or medication/substance intake.

The success of these techniques was highly heterogeneous. The predefined categories of using earplugs/earphones, closing/opening windows, and changing the bed position were rated as unsuccessful by the majority of respondents (between 72% and 83%) with elimination/switching off a suspected source being reported as an unsuccessful technique by the most participants (n = 123, 86%). The individually mentioned techniques tried by a smaller number of participants (<10) showed higher success proportions; especially medication/substance intake (100%, n = 3), coping (100%, n = 4), noise-canceling headphones (86%, n = 7), placing something in or on ears (86%, n = 7), distraction (80%, n = 5) or masking by ventilation (70%, n = 10). Notably, the success of some techniques was mentioned alongside a drawback, such as loud music being able to mask LFN, but still making it difficult for the participant to fall asleep. For more details, please refer to Appendix A.

### 3.2. LFN-Related Complaints

#### 3.2.1. Frequency of Complaints

On average, participants reported experiencing nuisance or hindrance due to LFN often (*M* = 3.1; *Mdn* = 3.0; *SD* = 0.9; n = 178). Most individuals experienced complaints continuously (36%) or often (38%), and the remainder experienced complaints regularly (14%), sometimes (5%), or never (1%). For an overview, see Appendix A. Participants from the comparison group that experienced nuisance regularly, often, or never were excluded, and the remainder consisted of individuals experiencing LFN-related hindrance sometimes (n = 152, 41%) or never (n = 219, 59%). Additionally to LFN-specific complaints, the comparison group also reported information on experienced nuisance from regular, non-LFN specific noise. An average of sometimes was reported (*M* = 0.9; *Mdn* = 1.0; *SD* = 0.7; n = 371) with most individuals experiencing complaints sometimes (63%), followed by never (26%), regularly (8%), often (3%), and continuously (0.3%).

#### 3.2.2. Extent of Complaints

On a scale from 1 (not at all) to 10 (very much), individuals reported the extent of LFN complaints in their daily living on average as a 7.1 (*Mdn* = 8.0; *SD* = 2.4; n = 185). Most frequently, participants reported LFN to have a high impact on daily living (scale scores 7:17%; 8:26%; 9:14%; 10:14%), and only a few participants reported a mediocre or low impact (scale scores 1:3%; 2:5%; 3:4%; 4:5%; 5:4%; 6:6%). For an overview, please refer to Figure 2.

Participants from the comparison group reporting the extent of LFN complaints in their daily living as a three or higher were excluded, the remainder reported an impact of 1 (n = 262, 71%) or 2 (n = 109, 29%). Regarding the extent of complaints from regular noise, the comparison participants reported an average of 2.0 (*Mdn* = 2.0; *SD* = 1.4; n = 371). Most frequently, participants in the comparison group reported regular noise to have a very low impact on daily living (scale scores 1:47%; 2:30%; 3:12%) and few reported a mediocre or high impact (scale scores: 4:4%; 5:2%: 6:2%; 7:2%; 8:0.5%).

#### 3.2.3. Physical and Psychological Complaints

Almost all participants (n = 187, 98%) reported at least one physical or psychological complaint attributed to experiencing LFN as depicted in Table 2. The number of different reported complaints showed a roughly normal distribution around the mean of 9.33 (*Mdn* = 9.0, *SD* = 4.7) and encompassed between one and 21 complaints, and one person naming 30 different complaints (Appendix A). Most frequently, participants reported difficulties with sleeping (90%) and fatigue (75%). Other complaints experienced by more than half of the participants were annoyance (63%), restlessness (63%), a pressing/pulsing in or on the ears (63%), stress (61%), concentration difficulties (60%), vibrations in their body (59%), and irritability (57%). Furthermore, many other complaints were experienced by a high number of participants, and various individual other bodily (23%), psychological (14%), or unclear physical or psychological (5%) symptoms were named.

#### 3.2.4. Social and Societal Consequences

More than half of the participants (n = 114, 60%) reported to experience at least one social or societal consequence from LFN as depicted in Table 3. Among the participants reporting complaints, between one and five different complaints with a mean of 1.7 (*Mdn* = 1.0, *SD* = 0.99) were named with most participants experiencing difficulties in one (n = 68, 36%) or two (n = 29, 15%) societal areas of their life (Appendix A). Most often, participants reported housing problems or the intention to move (27%), followed by relationship/family difficulties (23%) and work-incapacity/work-related problems (20%). Notably, the extent of the experienced impact varies greatly. Descriptions of social and societal consequences range from life adjustments such as trying to sit outside with visitors, abandoning one’s hobby, or being unable to work, to requesting euthanasia.

#### 3.2.5. Authorities LFN Was Reported to

Most participants (n = 174, 92%) reported the perceived LFN to at least one authority. From the participants that contacted an authority, between one and nine different authorities were contacted with a mean of 3.0 (*Mdn* = 3, *SD* = 1.6). Somewhat more than 20% reported one, two, three, or four contacted authorities, respectively, making up 87% of the respondents (Appendix A). Most participants contacted the municipality (n = 132, 70%), the Dutch Municipal Health Service (n = 108, 57%), or the Provincial Environmental Service (n = 85, 45%). Other reports were addressed to various other authorities related to health, environment, housing, the assumed source of the sound, the law, the public, or one’s community. For more details, please refer to Appendix A.

#### 3.2.6. Experts Consulted concerning Experienced Complaints

The majority of participants (n = 152, 80%) consulted an expert regarding their LFN-related complaints. Specifically, among participants that approached an expert, between one and nine different experts were reported with a mean of 2.6 (*Mdn* = 2.0, *SD* = 1.7). Most frequently one (35%), two (22%), or three (20%) experts were consulted (Appendix A). The most commonly consulted expert was the general practitioner (n = 130, 68%), though also audiologists (n = 68, 36%), ear, nose, and throat specialists (n = 61, 32%), psychologists (n = 41, 22%), neurologists (n = 20, 11%), and other experts were reported to have been consulted. For more details, please refer to Appendix A.

#### 3.2.7. Medication Usage

About 57% of the LFN group (n = 109) reported using medication including prescription-free medication (vitamins, food supplements, contraceptives, or alternative forms of therapy were not taken into account). Specifically, 23% reported to take at least one calming medication, 23% to take at least one cardiovascular medication, and 40% to take at least one other type of medication. In comparison, significantly less (X2 (1, N = 372) = 11.83, *p* < 0.001) individuals in the comparison group reported to take medication (n = 304, 42%). Although a similar proportion reported taking cardiovascular medication (24%), significantly fewer (X2 (1, N = 372) = 57.90, *p* < 0.001) individuals reported taking calming medication (3%) and another type of medication (25%, (X2 (1, N = 372) = 10.38, *p* < 0.001).

Regarding the number of different medications taken, the LFN participants reported to take between one and 10 different medications (*M* = 2.6, *Mdn* = 2.0, *SD* = 1.8) in contrast to the comparison group taking on average significantly less (U = 6687.0, *p* = 0.003) medication (*M* = 2.0, *Mdn* = 1.0, *SD* = 1.5, range: 1–8). Regarding the number of medication types taken, the LFN participants reported to take between one and six types of medication (*M* = 1.9, *Mdn* = 2.0, *SD* = 1.2) in contrast to the comparison group taking on average significantly less (*U* = 6726.0, *p* = 0.005) types of medication (*M* = 1.5, *Mdn* = 1.0, *SD* = 0.9, range: 1–6). For more details, please refer to Appendix A.

### 3.3. Characteristics of Individuals Perceiving LFN

#### 3.3.1. Age, Sex, Education, and Marital Status

Demographic characteristics of the LFN and comparison group are presented in Table 4. In terms of sex, the LFN group and CG showed a similar sex distribution. However, both groups presented with more females compared to the Dutch adult population. Considering age, the LFN group and CG showed a similar age average and distribution, although age was approximately normally distributed in the LFN group, and somewhat left skewed in the CG group. Compared to the Dutch adult population, individuals in both groups were on average about 10 years older. Regarding education, the LFN group and CG showed a similar distribution of middle and high-educated individuals, yet the CG presented with significantly less low-educated individuals with a small to medium effect size.

While the Dutch population (15 years and older) presents with an approximately equal distribution between the three education categories, the LFN group and CG consisted of two-thirds highly educated individuals and very few low educated individuals. Finally, the LFN group, the CG, and the Dutch adult population showed a similar distribution in marital status. Some minor differences included fewer unmarried individuals in the CG and more unmarried individuals in the Dutch population compared to the LFN group, fewer divorced individuals in the Dutch population compared to the other two groups, and fewer widowed individuals in the LFN group compared to the other two groups.

#### 3.3.2. Occupational Status

The occupational status of both groups is presented in Table 5. Significantly fewer individuals worked in full-time positions in the LFN group compared to the CG, though they worked significantly more hours. Further, significantly more individuals in the LFN group reported working part-time compared to the CG. The CG presented with comparable proportions to the work-eligible Dutch population (between 15 and 75 years). However, all group differences regarding full-or part-time work were of small effect size. There was no statistical difference found in the number of individuals being unemployed or years being unemployed between the LFN group and CG with a small effect size. Notably, there were slightly more individuals in the LFN group unemployed compared to the CG and even more to the Dutch work-eligible population, with two individuals in the LFN group being unemployed for over 30 years. Further, significantly more individuals in the LFN group were unable to work by being incapacitated or on sick leave with small to medium effect sizes compared to the CG. The Dutch work-eligible population showed a comparable proportion of individuals unable to work compared to the CG. No statistical difference was found in terms of the percentage of incapacity and sick leave, despite the CG showing a higher percentage of incapacity with a medium effect size. Finally, no significant difference was observed in the proportion of individuals being homemakers, students, and individuals in (pre-)pension. The Dutch work-eligible population entailed somewhat more students and less individuals in pension compared to the LFN group and distinctly less individuals in pension compared to the CG. Notably, significantly more individuals in the LFN group described volunteer work as an additional occupational status with a small to medium effect size.

#### 3.3.3. Field of Occupation

A current occupation was reported by 69% of the LFN group and 57% of the comparison group. This difference was significant with a small effect size (Appendix A). The remainder consisted of persons ineligible for work or whose daily living status is best described with something other than work including pension, student, or homemaker. Participants’ field of occupation is presented in Appendix A. All three groups performed mainly within the occupational fields of care and welfare (LFN:25%, CG:17%, DP:15%) and business and administrative professions (LFN:15%, CG:27%, DP:19%); and in the occupational fields of technical occupations in the LFN and DP group (LFN:19%, DP:14%) or educational professions in the CG (15%). Overall, the distribution of proportions in occupational categories differed between the three groups for most fields. The differences between the LFN and CG were, however, not significant in most cases, and only business and administrative professions were performed significantly less by the LFN group (*X*^2^ (1, *N* = 341) = 5.78, *p* = 0.016), and technical occupations significantly more often by the LFN group (*X*^2^ (1, *N* = 341) = 4.52, *p* = 0.034) with small effect sizes.

#### 3.3.4. Living Location

The distribution of individuals living in the Netherlands (LFN: n = 188, 99%; CG: n = 364, 98%) per province is presented in Appendix A and a geographical distribution of participants’ living places based on postal codes in Appendix A. Altogether, participants from all 12 provinces spread across the Netherlands were represented in the LFN group and the CG, and their distribution was roughly comparable to each other and the distribution of the Dutch population. There were four significant differences with small effect sizes in the distributions, such as more LFN participants living in the province Groningen, where the University conducting the research is also seated. A geographical depiction of the urbanization level of participants’ living location is provided in Appendix A. About 27% of the LFN participants live in extremely urbanized regions, 21% in strongly urbanized regions, and the remaining participants spread roughly equally over moderately, hardly, and unurbanized regions. This distribution is comparable to the Dutch population (Appendix A). The urbanization level of the LFN participants showed a low, nonsignificant, positive correlation with the frequency of experienced complaints shown (r (175) = 0.03, *p* = 0.70), and a low, nonsignificant, negative correlation with the extent of experienced complaints (r (181) = −0.06, *p* = 0.45). The geographical depiction of the frequency and extent of experienced LFN complaints does not show a clear pattern depending on participants’ living location as can be seen in Appendix A. Notably, there are even big differences in the perceived frequency and extent of complaints between individuals that live in close vicinity to each other as is the case in urbanized regions.

#### 3.3.5. Living Situation

A detailed overview of the participants’ living situation is depicted in Table 6. The proportion of participants living in different dwelling types did not differ significantly between the LFN group and the CG. Somewhat more than half of the individuals live in attached houses (58%), entailing semi-detached and terraced houses, and 20% live in flats, and 22% in detached houses.

The proportions of the Dutch population are overall similar, however with somewhat less individuals living in flats (18%) and detached houses (18%), and more living in attached houses (62%). Further, the LFN group lived for significantly fewer years in their current dwelling (*M* = 12.0, *Mdn* = 13) compared to the CG (*M* = 20.15, *Mdn* = 19), although both groups showed a similarly right-skewed distribution of years. Finally, the proportions of the household size were similar between the LFN group and CG. About one-third of the participants lives alone (33%), somewhat more than one-third lives with one other person (35%), and somewhat less than one-third with two or more persons (31%). In comparison, the Dutch population presented with less individuals living alone (18%) and more co-living with two or more persons (51%).

## 4. Discussion

This research aimed to provide an extensive description of the (1) LFN perceptions, (2) LFN-related complaints, and (3) demographic characteristics of individuals reporting to experience LFN in contrast to individuals reporting not to experience LFN. Results of this study are meant to encourage further research and present information on LFN for authorities, stakeholders, healthcare providers, and affected individuals.

### 4.1. LFN Perceptions

Overall, the reported LFN perceptions varied highly between individuals and were dependent on different circumstances. Although LFN was mainly perceived auditorily by this sample, the majority of participants also reported feeling LFN in their body, which could include a great variety of different body parts. Concerning the perceived sound, a variety of sounds was reported, most commonly a humming sound. Notably, individuals often described multiple sounds and for half of the participants, the sound was also accompanied by vibrations. This is in line with previous research, which found LFN to be perceived predominantly auditorily [5,25] and being mostly described as a humming sound [5,7,25]. However, the current research observed higher proportions of also other experienced perceptions or types of sounds. Differential research by, for example, audiologists, acousticians, neurologists or behavioral scientists investigating subgroups based on LFN perceptions (i.e., sound vs. vibration, constant vs. nonconstant perception) could lead to insights regarding the possible source of perception, the perceptual processing of LFN complainants, and the possible relation between perceptions and complaints.

Concerning the location of perceptions, LFN was always perceived at inside locations, and for three quarters of the participants, at outside locations. This finding seems especially interesting given that most assumed sources are related to inside locations and given that usually people spend most of their time at inside locations. The area of LFN perception differed individually from one´s own home to very specific places or to everywhere. This finding supports the previous research, which suggested that LFN is more frequently perceived in inside locations, specifically more often in one’s home [5,7,24,25]. In the cases in which an external source was identified, it would be interesting to document the congruence between assumed and measured sources and which sources are found the most commonly. Research trying to differentiate subgroups based on location of reported LFN perceptions could support the search for the underlying external or internal causes of LFN perceptions. Further, about three quarters of the participants reported that their LFN perceptions fluctuate and depend on specific circumstances. The main named factors were, in particular, an increased LFN perception in evenings/nights, as also found in previous research [5,24,37] or in silence when lacking masking noises [37]. However, described circumstances could be differing greatly between participants. These circumstances correspond with sleeping difficulties being the most commonly named complaint, and with participants trying to mask sounds to alleviate their perceptions.

Although two thirds of the LFN group requested a measurement of the LFN, only one third reported that the sound could be measured or perceived by the measuring instance, and only a quarter of the participants could provide the frequency of the LFN. A similar tendency can be observed in other research. In the survey by Møller and Lydolf [25], a LFN measurement was conducted for only about 50% of the participants that filed a LFN complaint. Measurements reported in that survey and conducted by other studies investigating the source of complaints [38,39,40] could only partially find external or internal sources of LFN perceptions. Further, only 8% of the participants that complained to an authority from the survey by Møller and Lydolf [25] indicated a (partial) solution of their problem. These findings have partially been attributed to measurement difficulties (e.g., insufficient equipment), however various hypotheses have also been formulated towards the source of LFN perceptions including external and internal sources. The majority of the participants could not report a measured frequency in their dwellings and among those who could, most reported a frequency in the LFN range. However, some also reported sounds reaching into the infrasound range, with very few reaching into the frequency range above the Dutch cut-off definition. On the one hand, these findings highlight the need for the development of LFN measurement instruments and exposure assessment protocols that are not just sensitive to both LFN and infrasound, but are also cost-and time-efficient to administer. This could, in turn, also help further research investigating potential differences in LFN and infrasound-related symptoms. On the other hand, these findings emphasize that there is a high number of individuals with substantial complaints where no measurement has been conducted (yet) or no source has been identified (yet), but to whom still further attention has to be paid and support provided.

Regarding the source of the perceived LFN as assumed by the current sample, about two thirds of the sample expressed various assumptions. However, there seems to be no clear consensus with a large assortment of different and individual sources being reported. The most commonly named source was ventilation, named by only one fifth of the participants. This seems to reflect the commonly observed difficulty of identifying a source of LFN. Interestingly, a various amount of the commonly named sources were already known for more than 50 years for their potential to emit low-frequency sounds when considering, for example, the controlling of noise on buildings, including industrial and commercial operations, traffic, air-conditioning, TV/radio, or transformers, as described in [41]. It must be noted though, that many sources were not predefined in the questionnaire by the researchers and were named independently. Future research asking specifically for these sources might, therefore, show different findings. Interestingly, most descriptions referred to technical sources.

When looking at the private environment of participants, two thirds reported that there was another person also perceiving the LFN, most often the partner of the individual affected. Further research into who these other persons are, into whether these other persons perceive the same sound (by for example sound measurements), and assessments of the extent of perceived disturbance of these other persons could provide further insights into the underlying external or internal cause of the LFN perceptions and underlying perception processing. Moreover, social research into the experienced burden of the personal environment of LFN complainants would be informative.

Finally, a considerable increase in the occurrence of the first LFN perceptions was observed over the past years with about a third of the participants being able to indicate a specific day on which they first perceived the LFN, and another third specifying a specific month for the first LFN occurrence. Notably, it is unclear from the current data whether the rise in first LFN perceptions might be related to an increase of LFN sources, an increase in LFN-sensitivity in the population, an increase in awareness about LFN, a possible sampling bias of individuals with a more recent start of complaints being more motivated to contribute to research, or other reasons.

Overall, despite some overarching patterns observed in the type of LFN perceptions, experiences seem to be manifold, highly individual, and not necessarily constant over time. Additionally, it seems that the perception of LFN cannot be traced back to a source or a measured sound frequency in many cases. Considering that all participants report LFN perceptions at inside locations, paired with the rising pattern of first occurrences of LFN perception, there is an urgent need for attention to the complaints of this group. Supporting the findings of previous research, LFN perceptions seem to have complex, multifactorial causes and the group of LFN complainants seems to be highly heterogeneous. Future replication research verifying and extending the current state of knowledge as well as multidisciplinary research (i.e., including acousticians, audiologists, and behavioral and social scientists) exploring specific subgroups and environmental vs. personal factors of this population seem crucial.

### 4.2. LFN-Related Complaints

Almost all participants in the LFN group reported multiple LFN-related complaints. Most participants stated to experience complaints often or continuously and indicated that those complaints have a high impact on their daily living. Specifically, almost all participants reported troubles with sleeping and fatigue, but also many individuals named restlessness, stress, concentration difficulties, a pressing/pulsing sensation in/on the ear or body vibrations. Many more complaints were experienced by a high number of participants, however, various individual complaints were also reported. The experienced complaints are in line with complaints summarized by previous reviews and surveys [5,7,12,25,28,35,42], while the results of this study especially highlight sleep and fatigue as key complaints. Considering the crucial role of sleep on daily functioning and health, further medical research, i.e., by neurologists, psychiatrists or psychologists, into the type of sleep difficulties and alleviation of sleep problems is advised. Further, it seems crucial to understand the relation of sleeping problems with other mentioned complaints, psychiatric disorders, and daily functioning, for example through network analyses. Overall, these frequently mentioned complaints can affect an individual’s ability to work and to live an independent life, as well as result in further health related consequences (e.g., depression) with various costs that have to be carried by the affected individual, health insurances, tax payers, and governmental authorities. Further interest and research into the complaints and their alleviation seems to be therefore also in the interest of (the nonaffected) society. Indeed, about half of the participants reported social and societal consequences. Most commonly, these were related to housing problems, relationships and family difficulties, or work/study-related problems, as also observed in previous reviews [5,12,28]. These reported consequences stretch through the core areas of life. Further investigation of these consequences by social scientists are advised and considerations of possible support that could be offered by social, governmental, and medical providers.

A variety of actions to reduce the hindrance due to LFN were taken by almost all participants, mainly by trying to reduce the noise, turning off suspected sources, or by turning on masking sounds. The success of these often-tried actions seemed to be limited in contrast to individual actions, such as distraction or medication/substance intake, which were claimed as more successful. Some similar measures were reported in previous studies by Veldboom et al. [37] and Møller and Lydolf [25] (i.e., masking sound, medication, or earplugs), however, these were rated as less successful as it was the case in this study. Studies investigating the effectivity of masking sounds for LFN [37,43,44], a measure rated as the most successful one by Veldboom et al. [37], suggest that it can be indeed helpful for some individuals. However, not all sound types seem to be helpful, not all individuals seem to benefit from masking sounds, and Veldboom and colleagues suggest that learning to live with the sound seems to also be a meaningful factor. Interestingly, in the current study medication intake was only mentioned by very few participants with the purpose of reducing LFN-related hindrance as opposed to 17% taking medication in the study by Møller and Lydolf [25] and 48% by Veldboom et al. [37]. Still, when looking at medication intake in general, notably more individuals in the LFN sample of this study took medication compared to the comparison group. Additionally, the LFN sample took more different medications and more types of medications, especially in terms of calming medications, but also other medications. Accordingly, further pharmacological research into the specific types of medication, the proportions of medications with and without prescriptions, and the purpose of medications, as well as risk assessments of drug interactions would be advised.

Eventually, the majority of participants consulted one or more experts about their complaints, twice as many as reported in the study by Møller and Lydolf [25] (41%). Most often, a general practitioner, but also other medical and noise-related specialists, were contacted. Additionally, almost all participants reported the perceived LFN to one or more authorities, mainly to the municipality, the Dutch Municipal Health Service, or the Provincial Environmental Service. This proportion was also higher than observed in the survey by Møller and Lydolf [25] (65%). Considering the high number of different experts and authorities contacted, it would be useful to determine whom to contact with which concern, to discuss the LFN-related responsibilities and competences of experts and authorities, and to determine useful collaborations between involved parties. Further, an investigation of whether contacted experts and authorities are knowledgeable about LFN complaints and about how to react to them would also be beneficial together with the development of standardized LFN complaint assessment procedure (e.g., as done by [42] or [31]). 

In conclusion, the complaints reported and their consequences on daily living are numerous and diverse, and the perceived frequency and impact of those complaints is rated on average as high. This is accompanied by an increased use of medication, multiple expert consultations, and reports made to authorities. All of these factors represent not only a psychological burden on affected individuals and their environment, but also represent a financial burden on the public. However, the underlying mechanisms and interactions and associations between complaints are not clear yet and further systematic research in medical, neurological, behavioral, and social sciences would be important with the use of standardized and validated measurement instruments. In terms of symptom alleviation, first research investigating the usefulness of psychotherapy [45], cognitive behavioral therapy [46], or relaxation therapy [44] suggest that these techniques can help improve coping and quality of life to some extent for some individuals. Further research with RCT’s, with larger samples sizes, and with the involvement of multiple influential factors and possible confounders is advised. Additionally, an investigation of differential LFN perceptions and coping strategies between individuals highly affected and little affected by LFN perceptions could yield useful insights into symptom reduction.

### 4.3. Characteristics of Individuals Perceiving LFN

In terms of demographic characteristics, the LFN group presented with older individuals, more females, more highly and less low educated individuals compared to the general Dutch adult population. These observations regarding age and sex differences fit with the distributions observed in previous surveys (i.e., [5,7,24,30,31]). However, it is not clear whether this observed population might experience LFN perceptions more often, whether it is more likely to report LFN-related complaints, or whether it is more likely to participate in research. Notably, the ability to perceive higher frequencies decreases with rising age (e.g., [47,48,49,50]), therefore the higher age observed in LFN complainants could be related to individuals with higher frequency hearing loss increasing their attention towards lower frequencies. However, the extent of hearing loss occurs differentially per frequency, per age (group), and by sex. Further research by audiologists and other hearing related experts investigating the (change in) hearing thresholds of individuals with LFN perceptions could provide insights into whether age and sex-related hearing changes might be related to the demographic profile of this population. However, at this very moment, inferences in terms of external validity have to be done carefully about the demographic data of this population. 

Considering participants’ occupational status, fewer individuals in the LFN group worked full-time and more worked part-time or were unable to work because of incapacity or sick-leave compared to both the Dutch population and a comparison group with similar age, sex, and education characteristics. In contrast to that, when categorizing individuals as having a current occupation (currently working or work eligible individuals) or not having a current occupation (persons not eligible for work or whose daily living status is best described with something other than work), more individuals in the LFN sample had a current occupational field compared to the comparison group. However, it has to be noted that there is a high proportion of work-eligible individuals in the LFN group that are currently unemployed, incapacitated, or on sick-leave. This observation is in accordance with the reports of participants describing work-related difficulties or job loss associated with their LFN perceptions. Further, more individuals in the LFN sample were in pension or pre-pension compared to the Dutch population. A possible reason for that could relate to the higher age average found in the LFN group with older individuals being more likely to perceive or suffer from LFN, or to individuals with LFN complaints or individuals going into pension earlier. However, there are even more individuals in the comparison group in pension, so another reason might be that individuals in pension have more time participating in research in general. In this regard, further sociological research into the association between occupational status and LFN specifically would be needed. Eventually, although individuals in the LFN group worked in different occupational fields and also mainly in the fields carried out by the majority of the Dutch population, there were noticeably more individuals working in care and welfare professions or technical professions. It would be interesting to explore the meaningfulness of these findings in subsequent research. Considering that the assumed sources of LFN mentioned in this study were predominantly technical appliances, it could be that individuals in these professions might be exposed to more LFN. Alternatively, it could be that individuals in these professions are more aware of perceptions and complaints and therefore are more likely to report their experiences. For example, qualitative interviews could investigate not only the associations between LFN and the presence of (assumed) LFN sources, and the possible differences in LFN perceptions at the workplace and at home, but also the role of work-related stress on LFN-related complaints. It must be noted that much of the research investigating the effect of LFN (exposure) on task performance has been done on convenience samples of high school/university students, i.e., refs. [51,52,53,54,55,56] and volunteers outside of an educational setting [57,58]. These samples often did not report LFN complaints in their daily living. Occupational research would often refer to office spaces or conditions designed to resemble office working conditions [53,54,59]; one study specifically included airline technicians [60]. Considering the demographic profile of LFN complainant samples and the occupational distribution observed in this research, future research could extend to investigate the role of LFN in other occupational settings, specifically in technical and healthcare professions. Concluding, the results of this study suggest that reported LFN perceptions could be associated with reduced work-force and relate to some occupational fields, which could have implications for legislators or occupational health and safety.

Finally, regarding the participants’ living location, individuals from the whole Netherlands were represented with an approximately same distribution between Dutch provinces in all groups. Interestingly, neither the urbanization level of dwellings, nor the dwelling type, nor the household size differed notably between the groups, although the general Dutch population lives somewhat more often in houses and less often in flats, with less individuals living alone. However, it also has to be considered that the comparison data of the Dutch population for occupation and living location referred to the whole Dutch population, including individuals under 18, and is therefore not a precise comparison group. Interestingly, Vasudevan and Gordon [24] found LFN to be mainly perceived in quiet rural and suburban areas, which might not only be related to location dependent noise exposure, but also to personal variables, such as higher expectations for quiet among rural residents (e.g., [61]). However, this was not found in the current study, which suggests, in contrast, that participants´ living place and urbanization level do not seem to be related to the frequency or extent of LFN complaints. LFN perceptions could rather depend on other factors and could have multilayered causes. This finding is especially interesting since the most assumed LFN-sources referred to technical appliances. This could raise the presumption that more individuals with LFN perceptions live in urbanized rather than rural areas. On the other hand, numerous participants described the LFN to be more prominent at night times and in silence, and that other sounds can mask the perception of LFN. Based on these descriptions, more LFN individuals could be living in rather silent, rural areas. Subsequently, future causal research could investigate the LFN sources, the expectations towards quiet/noise, and the differences in LFN complaints based on different urbanization levels. Further, research utilizing computer-based monitoring systems and geo-information systems such as described by Fidell [16] could also provide further insights into noise annoyance responses. Finally, participants of the LFN group live significantly fewer years in their current dwelling compared to the comparison group. This observation is in line with a quarter of participants reporting housing problems or the intention to move due to LFN-related complaints. Further research would be useful to investigate the frequency and reasons of moving in association with LFN perceptions.

### 4.4. Limitations

The restrictions of the recruitment methodologies for all groups pose one of the main limitations of the study and might limit the representativeness of the samples. The LFN sample, in contrast to various previous studies, aimed to consist of individuals in the general population without specifications of living or working locations associated with LFN exposure. Indeed, individuals across various demographic, occupational, and living contexts were recruited with differing extents of perceived nuisance from LFN. Still, it has to be noted that individuals were recruited mainly through a volunteer organization and may therefore encompass individuals with a higher perceived burden from LFN-related complaints and individuals highly motivated to share their experiences and to participate in research. For instance, the high number of complaints, medication taken, and experts and authorities consulted may be amplified in this research by this sampling method. Further, the high rate of first LFN perceptions occurring in the most recent years might be related to the individuals with the most recent start of LFN-related complaints being especially engaged with the topic and motivated to participate. Further, this sample could encompass more individuals with time to participate in research and the lower rate of work time in this sample could relate to individuals with less working hours being more likely to have time for research participation. Consequently, this recruitment context may encompass individuals with higher perceived burden from LFN. The true population of individuals with reported LFN perceptions might additionally encompass individuals with less LFN complaints, but also individuals with less time or opportunities to participate in research.

Concerning the comparison group, there might be a sampling bias present through the recruiting of an online research panel. Participants might have a heightened interest and more time to participate in research, which might be a possible explanation for the higher proportion of individuals being in pension in this research group. Finally, the comparison group of the Dutch population consisted of an adult population only for the analysis of age, sex, and marital status. Comparison data for education and occupational status was composed of specific subpopulations, i.e., including individuals aged 15 to 18, and included the full Dutch population for individuals’ living location and context. Comparisons with the Dutch population group must therefore be done carefully.

Further, the external validity of the LFN and comparison populations has to be treated very carefully. This study encompassed a Dutch sample and, in addition to the factors described above, might also add to a particular sample bias whereby behavioral research relies too heavily on samples of western, educated, industrialized, rich, and democratic societies [62].

Another major limitation refers to the subjective nature of the study. This research focused on the exploration of the subjective perception of LFN and related complaints in order to gain information for future research and for providing information for interested instances and affected individuals. Benefits of this study design include avoiding costly and time-consuming measurements or difficulties with using measurements that are sensitive enough to detect LFN components. However, since no objective measurement was conducted, this study design does not allow for causal conclusions between LFN exposure and subjective LFN perceptions. This means that this research cannot directly link noise exposure variables to the outcome variables and that some participants might experience complaints from sources other than LFN. Additionally, this study is also unable to differentiate between individuals affected by infrasound, LFN, or other lower medium frequency sounds. Accordingly, further experimental research is recommended, which aims to connect (1) objectively measured noise exposure variables, (2) different outdoor pseudo indicators (such as proximity to LFN noise sources), (3) indoor pseudo indicators (dwelling building materials), and (4) types of sounds (Infrasound, LFN, low medium frequency sounds) to the subjective experiences and complaints highlighted in this research. On the other hand, as previously described, studies investigating the source of complaints could only partially find external or internal sources of LFN perceptions and do not allow for a consensus. Correspondingly, it is similarly important to pay further attention to the experiences, complaints, and coping strategies also independent of the identification of a source.

Adding to that, it has to be considered that many of the above-described constructs are intertwined and likely to influence each other. For example, participants reported to always have LFN perceptions in inside locations. This could, for example, relate to LFN perceptions being more often present at night times, a time spent at home. It could also relate to the higher first occurrence of LFN in colder months compared to warmer months. This should be considered when interpreting the results of this study, when designing future research, or when experts and authorities are consulted about complaints. Furthermore, when considering the relation between relevant constructs, it has to be noted that noise-sensitivity has not been investigated in this research. However, noise-sensitivity represents a relevant non-acoustic factor that can influence noise perceptions and responses [11,14,17,20,21].

Finally, some of the results were based on answer choices predefined by the researchers, while some results are based on freely provided open answers or were based on additional remarks. Replication studies also providing these freely given results as answer options would be advisable, providing a more systematic investigation of those findings.

## 5. Conclusions

This study provided a detailed overview of LFN perceptions, LFN-related complaints, and the demographic characteristics of a Dutch sample reporting to experience LFN in comparison to individuals reporting not to experience LFN. Although this research supported some previous findings and identified some common perception and complaint patterns, it also highlights the individual nature of LFN-related experiences and the heterogeneity of this group. Perceptions seem to have complex and multifactorial causes that are not fully understood yet, and complaints seem intertwined with each other. In order to understand LFN complaints, an approach also considering physical, environmental and personal factors and their dynamic interactions seems necessary (e.g., the application of the bio-psycho-social International Classification of Functioning, Disability, Health (ICF), which views an individual’s functioning as the interaction between its health condition, environmental factors, and personal factors [63]). Furthermore, the results of this research suggest that affected individuals can undertake great efforts to understand their experiences and alleviate their complaints, amongst others through consulting various experts. However, it seems that the commonly observed uncertainty about the source of complaints can make it difficult for both, affected individuals and contacted authorities, to address the burden. The results of this research suggest that paying attention to the core complaints, especially sleep and fatigue, and their alleviation is crucial, and must also be independent of an established source. The LFN sample differed in regard to age, sex, and education from the general Dutch population and especially the role of age and sex might be interesting to consider in future research. Apart from this finding, no other LFN-specific distinct patterns in terms of demographic characteristics and lifestyle could be identified. This suggests that LFN-related complaints might be less dependent on these factors than on other personal factors. The findings of this research can hopefully provide new starting points for examining such factors while highlighting some more commonly observed factors. Despite numerous studies and efforts already addressing LFN-related perceptions and complaints, specifically more systematic, explorative, replicative, correlational, experimental, and hypothesis-testing research is advised using standardized and validated measuring instruments. Further, paying attention to the experiences of affected individuals, providing information to LFN complainants and concerned authorities, as well as the collaborations of multiple disciplines and authorities would be beneficial. These include physical, social, and behavioral scientists, audiological experts (e.g., acousticians, audiologists), healthcare experts (e.g., neurologists, psychiatrists, psychologists), health agencies (e.g., health insurance, municipal/governmental health services), environmental agencies (private or governmental environmental services), and also governmental authorities and the law (social security law, occupational safety). To conclude, it is evident that affected individuals are suffering and it seems that we currently cannot provide relief or remedy, as we do not understand the fundaments of their condition.

## Figures and Tables

**Figure 1 ijerph-20-03916-f001:**
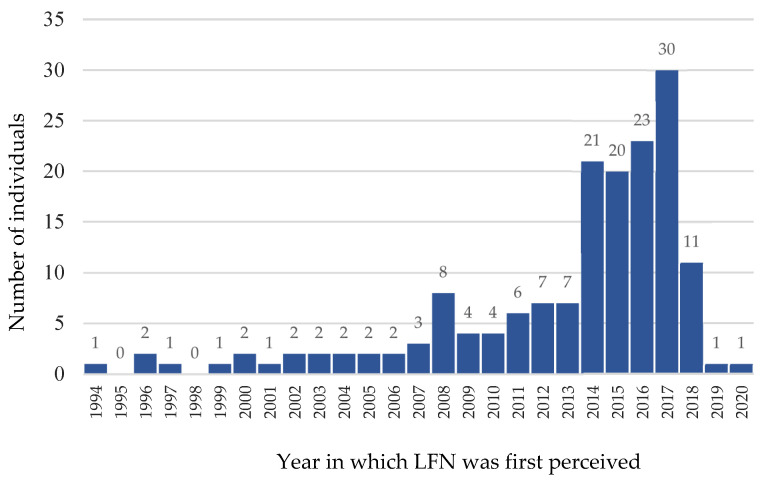
Year of first low-frequency noise (LFN) perception. Note: Data available from n = 178. Eleven did not provide a date, two did not know the first date, and one answer was noninterpretable. When LFN was starting to be perceived at different dates (n = 5), the first time it occurred was considered. Among those, three differentiated between a sound and vibration. Most data (n = 164) was collected in 2018.

**Figure 2 ijerph-20-03916-f002:**
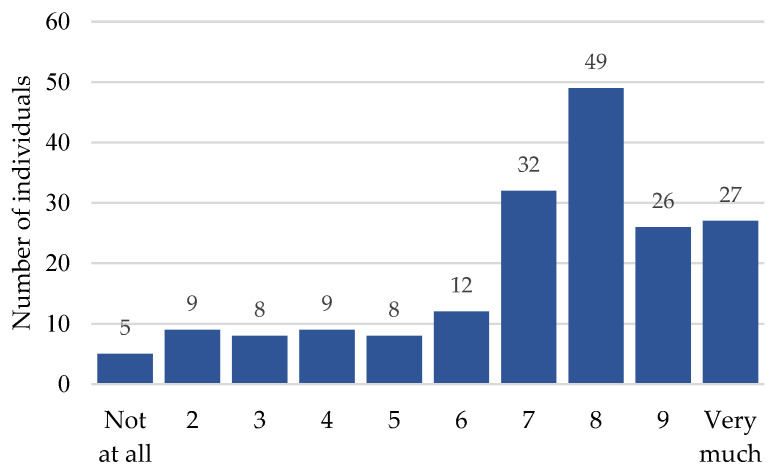
Extent of experienced complaints on daily life as indicated by the LFN group. Note: LFN group n = 190. Data was available from n = 185.

**Table 1 ijerph-20-03916-t001:** Measured constructs of the self-developed LFN questionnaire.

Construct Category	Construct (Type of Question)
In-exclusion criteria	Diagnosis of Tinnitus (CQ), diagnosis of neurological disorders (CQ/OQ), diagnosis of psychiatric or psychological disorders (CQ/OQ)
Demographic characteristics	Age (SA), sex (CQ), highest education (CQ/OQ), marital status (CQ/OQ), occupational status (CQ/OQ), occupation (SA), living location (SA), housing type (CQ), years spent in dwelling (SA), household size (CQ/OQ)
LFN perception	Date of first LFN perception (SA), type of sound (CQ/OQ), type of LFN perception (CQ/OQ), location of LFN perception (CQ/OQ), measurement of LFN (CQ), frequency of LFN (SA), assumed source (CQ/OQ), circumstances influencing LFN perception (CQ/OQ), other persons perceiving the LFN (CQ/OQ), actions, and their success to reduce nuisance from LFN (CQ/OQ)
LFN-related complaints	Frequency of complaints (CQ), extent of complaints (LS), physical complaints (CQ/OQ), psychological complaints (CQ/OQ), social and societal consequences (CQ/OQ), authorities LFN was reported to (CQ/OQ), expert consultation (CQ/OQ), medication use (SA)

Note: SA = short answer question. CQ = closed question. OQ = open question. CQ/OQ = mixture of closed and open questions where the participant can choose between options but might also add another option or comment to the question. LS = Likert scale question.

**Table 2 ijerph-20-03916-t002:** Reported LFN-related complaints.

Complaint (Multiple Answers Possible)	Frequency	Percent (n = 190)
Sleep difficulties	170	89.5
Fatigue	143	75.3
Annoyance	120	63.2
Restlessness	119	62.6
Pressing/pulsing in/on the ear	119	62.6
Stress	116	61.1
Concentration difficulties	113	59.5
Vibrations in the body	111	58.4
Irritability	109	57.4
Depressed Mood	88	46.3
Headache	79	41.6
Heart complaints	70	36.8
Pressure on the chest	61	32.1
Dizziness	53	27.9
Neck or back pain	48	25.3
Shortness of breath	43	22.6
Anxiety	40	21.1
Tightness of chest	23	12.1
Desperation/Powerlessness/Hopelessness	16	8.4
Reduced stamina/energy	11	5.8
Tension	7	3.7
High blood pressure	6	3.2
Gastrointestinal complaints	6	3.2
Forgetfulness	5	2.6
Hyperventilation	4	2.1
Anger/frustration	3	1.6
Other bodily symptoms ^a,b^	43	22.6
Other negative psychological feelings ^a,c^	26	13.7
Other symptoms (unclear physical or psychological) ^a,d^	10	5.3

Note: Underlined categories were predefined in the questionnaire. The originally separated questions of physical and psychological complaints were combined in this table due to the overlap between some of the categories. Data was available from n = 189. Since multiple answers were possible, the percentages do not represent the sum of the categories. ^a^ After data cleaning, 71 participants (37.4%) provided an open “other” answer. These individuals were re-categorized into a new system depicted in this table. The remaining “other” section refers to answers that were provided by two or less individuals and represent the number of different named complaints. ^b^ Includes, for example, sweating, high pulse, or whole-body pain. ^c^ Includes, for example, confusion, loss of control, or being tired of life. ^d^ Includes, for example, feeling hyperactive or falling ill more quickly.

**Table 3 ijerph-20-03916-t003:** Social/societal consequences.

Consequence ^a^ (Multiple Answers Possible)	Frequency	Percent (n = 190)
Housing problems/intention to move	52	27.4
Relationship/family difficulties	44	23.2
Incapacity for work/work-related problems	37	19.5
Social life problems	26	13.7
Job loss/Study stop	19	10.0
Life disruptions/adjustments	11	5.8

Note: Underlined categories were predefined in the questionnaire. Since multiple answers were possible, the percentages do not represent the sum of the categories. Data was available from n = 189. ^a^ After data cleaning 34 participants (17.9%) provided an open “other” answer. These individuals were re-categorized into a new system depicted in this table.

**Table 4 ijerph-20-03916-t004:** Demographic characteristics of the LFN group, comparison group, and Dutch population.

	LFN ^a^ (n = 190)	CG ^b^ (n = 371)	DP ^c^ (n = 17,475)	Comparison LFN and CG			
				χ^2^	*df*	*p*	*V*
Sex							
Females (%)	129 (67.9)	253 (68.2)	7174 (50.7)	0.01	1	0.94	>0.01
Education (%)				14.04	2	<0.001 ***	0.16
Low	14 (7.4)	5 (1.3)	4244 (24.3)	15.37	1	<0.001 ***	0.16
Middle	61 (32.3)	129 (34.8)	5229 (29.9)	0.45	1	0.50	0.03
High	114 (60.0)	237 (63.9)	4910 (28.1)	0.60	1	0.44	0.03
Marital status (%)				3.59	5	0.61	0.08
Married ^d^	92 (48.4)	188 (50.7)	6679 (47.2)	0.26	1	0.61	0.02
Unmarried	65 (34.2)	108 (29.1)	5248 (37.1)	1.53	1	0.22	0.05
No partner	34 (17.9)	55 (14.8)	-	0.89	1	0.35	0.04
Partner, living together	26 (13.7)	42 (11.3)	-	0.66	1	0.42	0.03
Partner, not living together	5 (2.6)	11 (3.0)	-	0.05	1	0.82	>0.01
Divorced	24 (12.6)	49 (13.2)	1378 (9.7)	0.04	1	0.85	>0.01
Widowed	7 (3.7)	25 (6.7)	860 (6.1)	2.18	1	0.14	0.06
	**Mean ± SD (Range)**			** *U* **		** *p* **	** *r* **
Age in years	57.6 ± 12.00 (18–87)	59.6 ± 11.7 (25–86)	49.6 ± 18.9 (18–109)	31,896.00		0.07	0.08

Note: Group comparisons were conducted between the LFN group and the CG. ^a^ LFN = LFN group. Data was available for sex and age from n = 190, for education from n = 189, and for marital status from n = 188. ^b^ CG = Comparison group. Data was available for sex, education, and age from n = 371 and for marital status from n = 370. ^c^ DP = Dutch population. Data from the CBS from 1 January 2021 was used for sex, marital status, and age from the Dutch adult population 18 years or older. For education, CBS data from the first quarter of 2021 for individuals 15 years and older was used. The number of individuals is provided in 1000 steps. ^d^ This includes marriage and registered partnership. % = Percentages from the total of the groups: LFN = 190, CG = 371, DP = 17,475,415 as of 1 January 2021. *** significant difference at a level *p* < 0.001.

**Table 5 ijerph-20-03916-t005:** Occupational status of the LFN group, comparison group, and Dutch population.

	LFN (n = 187)	CG (n = 371)	DP ^a^ (n = 17,475)	Comparison LFN and CG			
				χ^2^	*df*	*p* ^b^	*φ*
				*U*		*p*	*r*
Full-time (%) ^c^	31 (16.3)	102 (27.5)	47,420 (27.1)	8.2	1	0.004 **	−0.12
Hours per week;	42.7 ± 8.9 (34–65)	39.3 ± 5.94 (32–60)	39.7 (≥35)	1117.5		0.05 *	−0.08
M ± SD (Range) ^d^							
Part-time (%)	63 (33.7)	84 (22.6)	4430 (25.4)	7.8	1	<0.005 **	0.12
Hours per week;	24.1 ± 7.1 (3–32)	22.3 ± 6.9 (3–36)	21.2 (≤35)	1869.0		0.10	−0.14
M ± SD (Range) ^e^							
Unemployed (%)	11 (5.9)	13 (3.5)	445 (2.6)	1.7	1	0.19	0.06
Years unemployed ^f^	6.8 ± 10.5 (1–32)	4.3 ± 1.7 (2–7)	-	43.0		0.73	−0.09
Unable to work (%)	31(16.5)	11 (3.0)	785.500 (4.5)	33.10	1	<0.001 ***	0.25
Incapacitated	25 (13.4)	9 (2.4)	-	26.02	1	<0.001 ***	0.22
% Incapacitated ^g^	86.0 ± 18.1 (37–100)	94.4 ± 16.7 (50–100)	-	66.0		0.12	−0.31
Sick-leave	6 (3.2)	2 (0.5)	-			0.02*	0.11
% Sick-leave ^h^	100 ± 0 (100–100)	100 ± 0 (100–100)	-	5.00		1.00	0
Homemaker (%)	17 (9.1)	30 (8.1)	-	0.16	1	0.69	0.02
Student (%)	4 (2.1)	3 (0.8)	934 (5.3)			0.23	0.06
Pension/ Pre-pension (%)	52 (27.8)	133 (35.8)	3519 (20.1)	3.6	1	0.06	−0.08
Other (%)	20 (10.7)	6 (1.6)	-	23.1	1	<0.001 ***	0.20
Volunteer work	18 (9.6)	5 (1.3)	-	21.6	1	<0.001 ***	0.20
Individual responses	2 (1.1)	1 (0.3)	-	1.5	1	0.22	0.05

Note: Group comparisons were conducted between the LFN group and the CG. Since multiple answers were possible, the percentages do not represent the sum of all categories. ^a^ Data from the CBS was used. For full-time, part-time, and unemployment, data from the first quarter of 2021 were used based on the work eligible Dutch population between 15 and 75 years. For individuals unable to work and who are receiving pension, data from January 2021 was used based on the number of individuals receiving disability benefits (Dutch arbeidsongeschiktheidsuitkering) and the number of individuals receiving pension (Dutch AOW-uitkering). For students, data from the 2020/2021 academic year were used. The number of individuals is provided in 1000 steps. ^b^ Results are based on Fisher’s exact test for individuals on sick-leave and students. ^c^ Including employment and freelance work. ^d^ Data was available from LFN = 29 and CG = 100. ^e^ Data was available from LFN = 56 and CG = 80. ^f^ Data was available from LFN = 8 and CG = 12. ^g^ Data was available from LFN = 23 and CG = 9. ^h^ Data was available from LFN = 5 and CG = 2. % = Percentages from the total of the groups: LFN = 190, CG = 371, DP = 17,475,415 as of 1 January 2021. LFN = LFN group, CG = Comparison group, DP = Dutch population. * significant difference at a level *p* ≤ 0.05. ** significant difference at a level *p* < 0.01. *** significant difference at a level *p* < 0.001.

**Table 6 ijerph-20-03916-t006:** Living situation of the LFN group, comparison group, and the Dutch population.

	LFN (n = 190)	CG (n = 371)	DP ^a^ (n = 17,475)	Comparison LFN and CG			
				χ^2^	*df*	*p*	*φ*
Housing type (%) ^b^							
Flat	38 (20.0)	87 (23.5)	(17.5)	0.86	1	0.35	0.04
Attached House	110 (57.9)	208 (56.1)	(62.2)	0.17	1	0.68	−0.02
Semi-detached	35 (18.4)	63 (17.0)	-	0.18	1	0.67	−0.02
Terraced	75 (39.5)	145 (39.1)	-	<0.01	1	0.93	<−0.01
Mid-terrace	50 (26.3)	99 (26.7)	-	<0.01	1	0.93	<−0.01
End-terrace	25 (13.2)	46 (12.4)	-	0.07	1	0.80	−0.01
Detached house	42 (22.1)	76 (20.5)	(17.8)	0.20	1	0.66	−0.02
Household size (%) ^c^							
Living alone	63 (33.2)	123 (33.2)	3097 (17.7)	<0.01	1	0.93	<0.01
Living with others	125 (65.8)	248 (66.8)	14,120 (80.8)				
+1	67 (35.3)	133 (35.8)	5281 (30.2)	0.02	1	0.89	<0.01
+2	21 (11.1)	43 (11.6)	2822 (16.1)	0.04	1	0.85	<0.01
+3	23 (12.1)	56 (15.1)	3833 (21.9)	0.93	1	0.34	0.04
+≥4	14 (7.4)	16 (4.3)	2183 (12.5)	2.32	1	0.13	−0.06
	**Mean ± SD (Range)**			** *U* **		** *p* **	** *r* **
Years spent in dwelling	15.37 ± 12.0 (0–60)	20.15 ± 13.3 (1–63)	-	27,607.00		<0.001 ***	0.18

Note: Group comparisons were conducted between the LFN and CG group. Percentages from the total of the groups: LFN = 190, CG:371, and DP = 17,475,415 as of 1 January 2021. ^a^ For housing type, data from the EU statistics on income and living conditions (EU-SILC) was used from 2021. Only proportions in % were available from this data set. Proportions of the housing type do not add up to 100%, since only accommodation types asked from the LFN and CG group were used. For household size, data from the CBS from 2021 was used. Proportions of the household size do not add up to 100%, since this data only refers to private households and not individuals living in institutions. The number of individuals is provided in 1000 steps. ^b^ Attached house: a house that is attached to one or more dwellings/shares one or more common walls with another house. Semi-detached house: a house that is attached to only one more dwelling/shares one common wall with another house. Terraced house: a row of houses/similar houses linked together. Mid-terrace: a dwelling with other dwellings attached on both sides. End-terrace: the dwelling on the end of a terraced house row with only one dwelling attached to it. ^c^ Data available from LFN = 188. LFN = LFN group, CG = Comparison group, DP = Dutch population. *** significant difference at a level *p* < 0.001.

## Data Availability

An anonymized data set used for this study can be accessed from the corresponding author upon reasonable request.

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
