# Peer review of "Low-Frequency Noise: Experiences from a Low-Frequency Noise Perceiving Population"

_ijerph, 2023, doi:10.3390/ijerph20053916_

Round 1
Reviewer 1 Report
The manuscript ‘Low-Frequency Noise: Experiences from a Low Frequency 2 Noise perceiving population’ in its current form is not recommended for publication because of the following reasons:
1. The introductory section needs to be improved and clearly identify the research question/hypothesis. Once research question is established in Section 1, the statistical method(s) utilised need to be described in Section 2 and inferential statistics reported in Section 3.
2. The study considers a range of outcome variables, but it is unclear what exposure variables have been adopted. It is unusual to see a study regarding low frequency noise without a single mention of decibel, dB(A) and/or dB(C). In the absence of noise exposure variables, the authors may wish to consider outdoor pseudo indicators such as proximity to industrial areas, commercial areas, wind farms, airport, train stations etc as a way to define the potential variability in outdoor noise emissions and indoor pseudo indicators such as residential high-rise apartment living, within mixed-used buildings etc. as a way to define the potential for LFN from indoor sources (see A Guide to Airborne, Impact, and Structure Borne Noise: Control in Multifamily Dwellings).
3. It is unclear why the WHO Environmental Noise Guidelines for the European Region (2018), relevant systematic reviews and ICBEN studies are missing from the reference list. Page 47 of the WHO 2018 Guidelines highlighted the need to consider personal variables and situational factors as they can impact on noise annoyance judgements of residents. One of the key personal variables is ‘noise sensitivity’ (see https://doi.org/10.3390%2Fijerph13111163), and it is unclear why this variable is not evaluated by supplementary material 2 (administered questionnaire).
4. The authors have undertaken a cross-sectional study of low frequency noise experiences and presented a summary of survey results from a sample population 190 adults spanning across two countries (188 respondents from the Netherlands and 2 respondents from Belgium) without grouping respondents by location (e.g. urban, suburban, rural). Not only can exposure variable differ by location, personal variables such as higher expectation for quiet/peace may present amongst rural residents (see ISO 1996-1). Both exposure and personal variables linked to location can affect LFN response.
5. The authors should consider the paper by Sanford Fidell on "The Schultz curve 25 years later: A research perspective" which highlighted the geographic distribution of noise complaints can provide further insights into noise annoyance response which cannot be accounted for by the sole use of A-weighted noise exposure levels. In addition, Fidell noted in his paper that the increased interpretability of noise complaints made possible by computer-based record keeping and geoinformation system software suggests a more prominent role in the future for complaint rate information in the design of noise mitigation projects and impact assessments. Based on the above, the authors should consider presenting a geographical distribution of noise complaints with reference to surrounding noise sources.
Reviewer 2 Report
An interesting and extensive study. However, it should be further explained which LNF is emphasized in this study. Because below 20 Hz there is infrasound, which people cannot hear, but it has an effect on health and well-being. Above 20 Hz to 200 Hz the LF is already audible and the effect can be more easily examined. Since the subjects are not near the known LNF, it is difficult to say whether they are really affected by the LNF or the medium frequency sound.
Although the research is extensive, as the authors themselves point out, further research needs to be done in conjunction with sound source research and on-site analysis. The study shows that noise and LNF exist and is a big problem, it will help the researchers to organize further studies leading to a deeper analysis of the problem and clarification of the causes and solutions to solve the problem.
Round 2
Reviewer 1 Report
A key limitation of this study is the absence of noise exposure variables. I had previously raised this as an issue noting that the authors may wish to consider outdoor pseudo indicators such as proximity to industrial areas, commercial areas, wind farms, airport, train stations etc as a way to define the potential variability in outdoor noise emissions and indoor pseudo indicators such as residential high-rise apartment living, within mixed-used buildings etc. as a way to define the potential for LFN from indoor sources.
As the data around outdoor pseudo indicators is not readily available, it would appear this additional work is beyond the scope of this research at this point in time. As such, I have no other issues to raise and suggestions to offer. Grouping locations by the level of urbanization is not a substitute for outdoor pseudo indicators, and the authors have recognised this issue by discussing the need for outdoor pseudo indicators in Section 4.4 of the revised manuscript.
The data collected as part of this study and statistics reported in the manuscript provides useful information.